# Repetitive Compressive Loading Downregulates Mitochondria Function and Upregulates the Cartilage Matrix Degrading Enzyme MMP-13 Through the Coactivation of NAD-Dependent Sirtuin 1 and Runx2 in Osteoarthritic Chondrocytes

**DOI:** 10.3390/ijms26114967

**Published:** 2025-05-22

**Authors:** Masahiro Takemoto, Yodo Sugishita, Yuki Takahashi-Suzuki, Hiroto Fujiya, Hisateru Niki, Kazuo Yudoh

**Affiliations:** 1Department of Orthopaedic Surgery, St. Marianna University School of Medicine, Sugao 2-16-1, Miyamae-ku, Kawasaki 216-8512, Japan; masahiro.takemoto@marianna-u.ac.jp (M.T.); h2niki@marianna-u.ac.jp (H.N.); 2Department of Frontier Medicine, Institute of Medical Science, St. Marianna University School of Medicine, Sugao 2-16-1, Miyamae-ku, Kawasaki 216-8512, Japan; yodo@marianna-u.ac.jp (Y.S.); y2takahashi@marianna-u.ac.jp (Y.T.-S.); 3Department of Sports Medicine, St. Marianna University School of Medicine, Sugao 2-16-1, Miyamae-ku, Kawasaki 216-8511, Japan; fujiya-1487@marianna-u.ac.jp

**Keywords:** osteoarthritis, chondrocyte, repetitive mechanical stress, matrix metalloprotease (MMP)-13, runt-related transcription factor (Runx) 2, sirtuin 1, nicotinamide adenine dinucleotide (NAD), mitochondria activity

## Abstract

Mechanical stress is known to be a pivotal risk factor in the development of OA. However, the involvement of repetitive compressive loading in mitochondrial dysfunction in chondrocytes remains unclear. The aim of this study was to investigate whether physiologic levels of repetitive mechanical force affect the regulation of energy metabolism and activities of mitochondrial function regulators, sirtuin 1 and nicotinamide adenine dinucleotide (NAD) in chondrocytes, and to clarify any correlation with chondrocyte catabolic activity. Repetitive physiological mechanical stress was applied in a 3D chondrocyte-collagen scaffold construct, and the 3D cultured tissues were collected at different time points by collagenase treatment to collect cellular proteins. Changes in chondrocyte activity (cell proliferation, MMP-13 production), energy metabolism regulator levels (sirtuin 1), mitochondrial function (ATP production, NAD level), and the expression level of the osteogenic and hypertrophic chondrogenic transcription factor, runt-related transcription factor 2 (Runx2), were measured. Treatment with repetitive compressive loading resulted in no significant change in the cell viability of chondrocytes. In the repetitive mechanical loading group, there were statistically significant increases in MMP-13 production and expression of both sirtuin 1 and Runx2 in chondrocytes relative to the non-loading control group. Furthermore, ATP production and NAD activity in mitochondria decreased in the repetitive mechanical loading group. Our present study reveals that in chondrocytes, repetitive compressive loading accelerated sirtuin activation, which requires and consumes NAD within mitochondria, leading to a decrease of NAD and ultimately in reduced mitochondrial ATP production. Additionally, since sirtuin 1 is known to positively regulate Runx2 activity in chondrocytes, the activation of sirtuin 1 by repetitive load stimulation may induce an increase in the expression of Runx2, which promotes the expression of MMP-13, and subsequently enhances MMP-13 production. Our findings indicate that repetitive compression loading-mediated mitochondrial dysfunction plays a pivotal role in the progression of OA, primarily by driving the downregulation of ATP production and promoting the expression of the matrix-degrading enzyme MMP-13.

## 1. Introduction

Osteoarthritis (OA) is a common chronic and degenerative joint disorder characterized by degradation of articular cartilage, synovial inflammation, osteophyte formation, and subchondral bone sclerosis, leading to significant disability, joint pain and reduced quality of life [1,2,3,4]. With the advance of the disease, secondary arthritis, such as such as synovitis, soft tissue inflammation is observed in several joint tissues. It has been also observed the degeneration and inflammation of soft part tissues, such as the infrapatellar fat pad, meniscus, and synovial tissues in patients with knee OA. In Japan, there are over 25 million OA patients over the age of 50 who have X-ray-detected joint deformities, and it is estimated that there are over 12.8 million OA patients who actually have symptoms such as joint pain and functional impairment [5,6,7]. In addition, there are over 900,000 new OA patients every year, and the economic loss has been estimated to be 1% of the gross national product, there are concerns that the number of patients will continue to increase in the future [8,9].

Several key risk factors, including age, genetics, gender, joint injury, obesity, and repetitive mechanical stress influence OA [10,11]. Aging remains the most significant factor, as cartilage naturally wears down over time. Genetic predispositions increase susceptibility, particularly in hand and knee OA. Obesity also accelerates OA in weight-bearing joints by increasing mechanical load and contributing inflammatory mediators. In addition, joint injuries and occupations involving repetitive motions further stress joints, leading to cartilage breakdown and secondary inflammation of soft part tissues including synovial tissue, infrapatellar fat pad, and meniscus. Along with that, metabolic disorders and low bone density also play a role, linking OA with systemic inflammation and joint degeneration [12,13,14]. During the development of OA, extrinsic abnormal stresses including mechanical loading, joint injury, obesity, and joint instability, or intrinsic stresses, such as aging, genomic/metabolic abnormality, and apoptosis, have the potential to change the normal activities of chondrocytes and induce homeostatic imbalance of articular cartilage [15,16]. Of several key risk factors for OA, recent insights have identified mechanical stress as being pivotal in OA pathogenesis, contributing to chondrocyte mitochondrial dysfunction, oxidative stress, apoptosis, and extracellular matrix degradation in articular cartilage [17,18,19,20].

To date, many studies have highlighted the role of elevated mitochondrial reactive oxygen species (ROS), that are induced by mechanical stress, in accelerating cartilage degeneration and contributing to OA development [21,22,23,24,25,26,27]. Previous reports have clearly demonstrated that mechanical stress secondarily induces the production of excess amounts of ROS by chondrocyte mitochondria, consequently, leading to further cell damage and cartilage matrix degeneration [24,25,26,27]. Green et al. have demonstrated that mechanical stress on cartilage tissue induces excessive production of ROS from chondrocytes and that the excess oxygen free radicals induce chronic catabolism of chondrocytes and cartilage matrix damage in articular cartilage tissue [23,26,27]. These findings clearly indicate that excessive ROS production in chondrocytes, driven by mitochondrial dysfunction in response to mechanical stress leads to oxidative stress and cellular damage, promoting chondrocyte apoptosis and arthritis within joint tissues. Mechanical stress-induced ROS production by mitochondria is thought to reduce the homeostasis of articular cartilage, and as a result, causes the progression of OA [25,26,27].

Furthermore, mitochondrial dysfunction gives rise to a reduction of adenosine triphosphate (ATP) production, thereby impairing chondrocyte function and undermining cartilage maintenance [28,29]. Mitochondria, often termed the “energy generator” of the cell, are dynamic organelles crucial for cellular energy metabolism. Mitochondria are not only energy generators but are also involved in several critical cellular metabolism processes, including the regulation of apoptosis, calcium homeostasis, as well as generation of ROS [28,29,30]. In general, chondrocyte mitochondrial dysfunction, and reduction of ATP production with a simultaneous increase in ROS generation, may play a significant role in the onset and progression of OA [31,32]. Thus, the study of the underlying mechanisms linking mechanical stress to mitochondrial dysfunction is important in order to clarify OA pathophysiology and to develop novel therapeutic strategies in OA.

Although mechanical stress is recognized as a catabolic factor for OA progression, physiological load stress is unavoidable for articular cartilage tissue, and there may be a mechanism for maintaining tissue homeostasis in the cellular response to physiological load stress. Indeed, our previous study revealed that the physiologic level of continuous loading resulted in no significant change in chondrocyte activity, suggesting that normal chondrocytes may have a stress tolerance against the physiologic level of continuous loading [33]. However, it remains unclear whether repetitive compressive loading influences mitochondrial function in chondrocytes. Even within the physiological load range, repetitive compressive loading to the articular cartilage may induce mitochondrial dysfunction in chondrocytes.

In OA, chondrocyte variability and the properties of cartilage matrix components may affect the resulting mechanical behavior of the joint. Recently, it has been demonstrated that the importance of chondrocyte and articular cartilage biomechanics with direct connection to their biochemical functions and activities [34]. Chondrocytes are responsible for the synthesis and degradation of the extracellular matrix/pericellular matrix. The pericellular matrix is considered to be a buffer for physical forces between the chondrocyte and the extracellular matrix. If chondrocytes are subjected to abnormal mechanical stimuli (excessive loading, joint trauma), their metabolism balance becomes altered, causing matrix loss and tissue degeneration, which can lead to OA. In chondrocytes, the cytoskeleton, the cilium, and calcium channels are the main subcellular components involved in the biomechanical response of the cells. The changes in mechanical properties of chondrocytes and cartilage matrix may affect their biomechanical behavior, resulting in the change of biomechanical behavior of the joint tissue itself. In pathological conditions of OA, chondrocytes and cartilage are subjected to several changes that also modify their biomechanical behavior. Indeed, OA chondrocytes showed a lower elastic modulus and viscosity compared to healthy chondrocytes. In addition, it has been also demonstrated that OA cartilage acquires a lower Young’s modulus compared to healthy cartilage [34].

Of the several regulatory factors involved in mitochondrial metabolism in response to mechanical stress, for this study we have focused on two key regulators of mitochondrial function: nicotinamide adenine dinucleotide (NAD) and sirtuin 1. Previous reports revealed that NAD is a coenzyme for cellular energy ATP production by mitochondria and that 30~70% of intracellular NAD is present within mitochondria [34,35,36]. It is well known that NAD is a master regulator of energy metabolism in mitochondria. Sirtuin 1 is a longevity gene-related NAD-dependent deacetylase, and for its activation, NAD is required as a coenzyme and consumed by mitochondria [37,38,39]. It has been indicated that sirtuin 1 also has an important role as an energy sensor during ATP production by mitochondria [40,41].

Recent studies indicate that sirtuin may have two important roles: response to stresses and regulation of energy metabolism which have been implicated in the pathophysiology and pathogenesis of a variety of diseases including OA [40,41,42]. NAD-dependent sirtuin activation is involved in stress responses by regulating mitochondria function, apoptosis, and cell metabolism through the deacetylation of target proteins in cells [40,43]. A better understanding of the involvement of mitochondria dysfunction, especially the NAD-sirtuin 1 pathway, in mechanical stress tolerance and energy metabolism, could lead to new insights into the development of novel diagnostic tools and therapies for OA.

Mechanical stress is known to be a significant risk factor in the development of OA and is involved in mitochondrial dysfunction, oxidative stress, and cartilage matrix degradation in OA [18,19,20]. However, the involvement of repetitive compressive loading in mitochondrial dysfunction in chondrocytes remains unclear. Clarifying the underlying mechanisms linking repetitive mechanical stress to mitochondrial dysfunction in chondrocytes could provide valuable insights into the relationship between mechanical stress and OA pathophysiology. Therefore, this study aims to investigate the effects of repetitive compressive loading on chondrocyte mitochondrial dysfunction in an OA in vitro model. The results of our study could provide important evidence to develop better therapeutic strategies against OA progression.

## 2. Results

### 2.1. Repetitive Compressive Loading of the 3D Cell Culture Construct

The 3D cell-collagen scaffolds were placed into individual wells of a 12-well culture dish and maintained in 1.5 mL DMEM. Then, repetitive compressive loading was applied to the 3D cell-collagen scaffolds with the use of a custom-designed and built apparatus as shown in Figure 1.

A physiologic and repetitive compressive loading was applied to the test 3D constructs at 0 kPa (non-loaded) or 25.5 gf/cm^2^ (2.5 kPa). Our preliminary experiments and previous reports confirmed that the physiological load does not induce cell damage or cell death (apoptosis), and so all subsequent experiments were carried out at 25.5 gf/cm^2^ (2.5 kPa) [26,44,45,46].

### 2.2. Effect of Physiologic and Repetitive Compressive Loading on Cell Protein Content in a 3D Cell-Culture Construct After Mechanical Loading

As shown in Figure 2A, repetitive compressive loading resulted in no significant differences in the cell protein content in the 3D cell-culture construct after mechanical loading, as compared to the non-loading control (*n* = 15, control vs. 15-min loading: *p* = 0.065, control vs. 60-min loading: *p* = 0.187, Figure 2A).

### 2.3. Effects of Physiologic and Repetitive Compressive Loading on Chondrocyte Viability

The effects of repetitive compressive loading on cell viability and cytotoxicity (cell membrane damage) in chondrocytes were analyzed by a Cell Counting Kit (CCK)-8 assay kit and a Cytotoxicity LDH Assay Kit-WST, respectively. Results demonstrated no significant difference in the proliferative potential of the cells in the mechanical loading groups and non-loading control (Figure 2B). These results indicated that the treatment of chondrocytes with physiologic and repetitive compressive loading did not affect the cell viability in comparison with the non-loading control.

The treatment of chondrocytes with repetitive compressive loading significantly induced cell membrane damage in comparison with the control (*n* = 4/each experiment, control vs. 15-min loading: *p* = 0.014, control vs. 60-min loading: *p* = 0.009, Figure 2C).

### 2.4. Scanning Electron Microscopy

As shown in the representative images in Figure 3, no significant difference was observed in chondrocyte number in the 3D cell-culture construct between the non-loading control (A) and the mechanical loading groups (B: 15-min loading) and (C: 60-min loading) [mean chondrocyte number/100 × 100 μm^2^ at x250 image (*n* = 10): (A) 6.2 ± 2.1, (B) 5.2 ± 1.8, (C): 5.9 ± 1.8, (A) vs. (B), *p* > 0.05, (A) vs. (C), *p* > 0.05]. In both groups, chondrocytes exhibited many protrusions and microgranules on the cell surface. The level of cell surface protrusions/granules was much higher in non-loaded chondrocytes than in mechanically 60-min loaded chondrocytes (*n* = 10, *p*< 0.001).

### 2.5. Effect of Physiologic and Repetitive Compressive Loading on Chondrocyte Activities, MMP-13 Production by Chondrocytes

As shown in Figure 4, repetitive compressive loading for 180 min, but not for 60 min, significantly upregulated the secretion of MMP-13 by 3D cultured chondrocytes as compared to the non-loading control (*n* = 4/each experiment, control vs. 60-min loading: *p* = 0.118, control vs. 180-min loading: *p* = 0.012).

### 2.6. Effects of Physiologic and Repetitive Compressive Loading on the Expression of Sirtuin 1 in Chondrocytes

Increased sirtuin 1 expression in chondrocytes was directly related to repetitive compressive loading time (*n* = 3/each experiment, Figure 5A,B). Treatment for 60 min with repetitive compressive loading significantly upregulated the expression of sirtuin 1 in chondrocytes, as compared to the control (*n* = 4 independent experiments, control vs. 15 min-loading: *p* = 0.164, control vs. 60 min-loading: *p* = 0.044, Figure 5C).

### 2.7. Effect of Physiologic and Repetitive Compressive Loading on the Level of NAD in Chondrocytes

As shown in Figure 6A, treatment with repetitive compressive loading caused a decrease in the levels of NAD activity in chondrocytes, compared with the control in each experiment (*n* = 4/each experiment). There was a significant difference in the NAD activity between the non-loading control and 60-min loading group (*n* = 4 independent experiments, control vs. 60-min-loading: *p* = 0.042, Figure 6B). In contrast to the 60-min loading group, no significant difference was observed between the 15-min loading group and the non-loading control (Appendix A).

### 2.8. Effects of Physiologic and Repetitive Compressive Loading on the Expression of Runx2 in Chondrocytes

The repetitive compressive loading groups showed upregulation of expression of Runx2 in chondrocytes in comparison to the non-loading control group in each experiment (*n* = 4/each experiment) (Figure 7A,B). There was a significant difference in the Runx2 expression in chondrocytes between the non-loading control and 60-min loading group (*n* = 4 independent experiments, control vs. 15 min-loading: *p* = 0.054, control vs. 60 min-loading: *p* = 0.002, Figure 7C).

### 2.9. Effect of Repetitive Compressive Loading on ATP Production by PBMCs

Treatment with the repetitive compressive loading induced to decrease in the levels of ATP production in chondrocytes, in comparison with the control (*n* = 4/each experiment, Figure 8A). The level of ATP production in the 60-min loading group showed a significant decrease compared with the non-loading control group (*n* = 4 independent experiments, control vs. 60 min-loading: *p* = 0.043, Figure 8B).

## 3. Discussion

The progressive OA is primarily marked by the degeneration of articular cartilage, alongside changes in subchondral bone structure and soft part tissues, such as infrapatellar fat pad and meniscus, synovial inflammation, and the formation of bony growths, or osteophytes. Several key risk factors, including age, genetics, gender, joint injury, obesity, and repetitive mechanical stress influence OA [16,18,20,47,48,49,50,51]. The various risk factors are involved in OA pathogenesis with varying importance. Indeed, mechanical stress is only one of these risk factors. Therefore, to clarify the exact mechanism of OA pathophysiology, it is also necessary to analyze the contribution of factors other than mechanical stress. In addition, since mechanical stress is thought to affect articular tissues with the exception of cartilage tissue, it is necessary to analyze the effects of mechanical stress on the activity of cells except chondrocytes. In the present study, first of all, we have studied the chondrocyte activity, to verify the involvement of repetitive mechanical stress in the OA pathophysiology. Next, we are planning to study the effects of repetitive mechanical stress on other cell activities in the joint tissue.

Many studies have provided evidence to support the role of elevated mitochondrial ROS produced by mechanical stress in accelerating cartilage degeneration and contributing to OA development [25,28]. Since mitochondria are not only ATP energy generators but also involved in ROS generation in response to mechanical stress, mitochondrial dysfunction could lead to both reduced ATP generation and ROS-induced mitochondrial oxidative stress, thereby impairing chondrocyte function and compromising cartilage maintenance [25,28,52,53]. With respect to mitochondrial metabolism, we focused on two key regulators of mitochondrial function, NAD+ and sirtuin 1. Generally, 30~70% of intracellular NAD+ is found in the mitochondria [36,37,38,39]. It Is well known that NAD+ has a role as a coenzyme In cellular energy ATP production by mitochondria and that it plays a role as a master regulator of energy metabolism in mitochondria [38]. As with NAD+, sirtuin 1, a longevity gene-related protein, also has an important role as a mitochondrial regulator acting as an energy sensor during ATP production by mitochondria [39,40,41]. Sirtuin protein is known to be a NAD-dependent deacetylase, and for its activation, NAD+ is required as a coenzyme [37,39]. Thus, along with sirtuin activation, NAD+ is consumed and the level of NAD+ in the mitochondria is decreased (Figure 9A).

In the present study, repetitive compressive loading induced upregulation of intracellular sirtuin 1 in chondrocytes. Since sirtuin 1 activation is known to require NAD+ from mitochondria, the level of NAD+ within mitochondria may decrease in response to mechanical loading. Indeed, our data showed a tendency for decreased levels of NAD+ in chondrocytes after treatment with repetitive compressive loading in vitro. As shown in Figure 9B as indicated by the blue arrow, we concluded that mechanical stress-induced sirtuin 1 activation leads to the consumption of NAD+ from mitochondria and a declining trend of NAD+ in mitochondria of chondrocytes. Subsequently, since NAD+ acts as a coenzyme in ATP production by mitochondria, there is no doubt that NAD+ reduction in mitochondria leads to decreased mitochondrial ATP production in chondrocytes. As expected, our results also indicated that repetitive compressive loading induced the reduction of ATP production in chondrocytes. These findings indicate that repetitive compressive loading induces the downregulation of ATP energy generation by mitochondria through a mechanism involving the sirtuin 1-NAD interaction in chondrocytes. Repetitive compressive loading may induce sirtuin activation, which consumes NAD in mitochondria, resulting in a decrease of NAD and ultimately in reduced ATP production in chondrocytes (Figure 9B as indicated in blue arrow).

NAD-dependent sirtuin activation is implicated in stress responses by regulating mitochondria function and cell metabolism through the deacetylation of target proteins in cells [42,43]. In the present study, repetitive compressive loading induced the upregulation of the intranuclear transcription factor Runx2, which is a master regulator for the hypertrophic differentiation of chondrocytes in osteoarthritic articular cartilage. In healthy articular cartilage, chondrocytes resist proliferation and terminal differentiation [1,16]. In contrast, chondrocytes in degenerated OA cartilage progressively proliferate and develop hypertrophy [16]. Runx2 is a key regulator of the transition from proliferating to hypertrophic chondrocytes in OA progression [54,55]. It is shown that Runx2-deficient mice lack hypertrophic chondrocytes in articular cartilage, suggesting the involvement of Runx2 in the de-differentiation of normal chondrocytes into hypertrophic chondrocytes in OA articular cartilage [55]. It is recognized that Runx2 is an important intranuclear transcription factor for the hypertrophic differentiation of chondrocytes during the progression of OA. Previous reports including our report have already demonstrated that sirtuin 1 positively regulates and accelerates the expression of intranuclear Runx2 in chondrocytes [33,56,57]. Our findings show that repetitive compressive loading induces both intracellular Runx2 and intranuclear sirtuin 1 expression in chondrocytes, suggesting that repetitive mechanical stress-induced sirtuin 1 activation leads to increased expression of Runx2, which in turn causes normal chondrocytes to de-differentiate into hypertrophic chondrocytes, resulting in the progression of OA. (Figure 9B as indicated by the red arrow).

Furthermore, numerous reports have shown that Runx2 promotes MMP-13 activation [58,59]. MMP-13 is an enzyme that degrades cartilage matrix [58]. Furthermore, it has also been demonstrated that hypertrophic chondrocytes highly express both MMP-13 and its positive regulator, the transcription factor Runx2 [60]. As well as Runx2, MMP-13 is a well-established marker for the hypertrophic stage of de-differentiation of normal chondrocytes into hypertrophic chondrocytes during the progression of OA [58,59,60]. There is a general consensus that MMP-13 is closely related to the pathogenesis and pathophysiology of OA [59]. In the present study, repetitive compressive loading significantly accelerated MMP-13 production by chondrocytes, as well as sirtuin 1 and Runx2 in chondrocytes. These findings indicate that repetitive mechanical stress-induced Runx2 expression leads to the increase of MMP-13 production by chondrocytes since Runx2 is known to be a promoter of MMP-13 in chondrocytes (Figure 9B as indicated by the red arrow). We, therefore, conclude that the repetitive mechanical loading-upregulated sirtuin 1-Runx2 pathway may lead to the acceleration of MMP-13 production by chondrocytes in OA.

The initial activation of sirtuin 1 in this NAD-dependent sirtuin 1-Runx2-MMP-13 pathway consumes NAD+ within mitochondria in chondrocytes. Therefore, repetitive compression loading also results in a decrease in ATP production by mitochondria as a result of the consumption of NAD+. The three sets of results from the different aspects of the pathway (ATP-downregulation, Runx2-mediated hypertrophic change of chondrocytes, MMP-13-upregulations) suggest an impact of repetitive compressive stress in the degradation of articular cartilage in OA.

Our in vitro study has indicated that the level of chondrocyte viability showed no significant difference between the repetitive compression loading group and the non-loading control group but, in contrast, the level of cell membrane damage was significantly higher in the repetitive compression loading group. These findings are consistent with the results of SEM. As shown in the SEM images in Figure 3, physiologic and repetitive mechanical loading decreased the level of cell surface protrusions/microgranules in chondrocytes, but not the cell number, suggesting cell membrane damage by repetitive compression loading. However, in the present study, the composition of protrusions or microgranules remains unknown. To verify and confirm the composition of microgranules, we plan to analyze the microgranules by immunostaining.

We have also studied the difference in the mechanical stress response to continuous/physiologic loading, repetitive loading, and long-term loading. In contrast to the current results, from a repetitive compression loading assay, our previous study clearly demonstrated that a physiologic level of continuous loading did not result in any significant changes in the activity of the sirtuin 1-Runx2 pathway and chondrocytes (cell growth, MMP-13 and proteoglycan productions) [33]. Articular cartilage is an avascular tissue that has a poor spontaneous self-healing capacity [1,16]. Adult chondrocytes in the cartilage tissue have little or no proliferation potential [16]. To maintain the function of articular cartilage in the joint, chondrocytes might have a mechanical stress tolerance to some degree. A better understanding of the involvement of mitochondria dysfunction, especially the NAD-sirtuin 1-Rux2 pathway, in mechanical stress tolerance and energy metabolism, could lead to new insights for the development of novel diagnostic tools and therapies for OA.

In the present study, regarding the production of MMP-13 by cultured chondrocytes, although there was a tendency to increase the secretion of MMP-13 enzymatic protein from chondrocytes in the 15-min and 60-min groups in comparison with the control, the results for both groups were not as significant as those for the 180-min group. We believed that MMP-13 protein synthesis/secretion by chondrocytes may require at least 60 min, but about 180 min. In contrast to the MMP-13 enzymatic protein, it has been suggested that changes in the expression and activity of both intranuclear Runx2 and sirtuin 1, which are regulatory factors for MMP-13 production by chondrocytes, and mitochondrial NAD may appear to occur within 60 min, which is shorter than the time required for MMP-13 protein production by the cells. Therefore, for the analysis of protein expression (MMP-13), a 180-min loading condition was added to the 15- and 60-min loading conditions and intracellular/nuclear transcription factor groups were evaluated under 15- and 60-min loading conditions. The difference in evaluation time depending on the factor examined was due to the consideration of the time lag in protein synthesis/secretion that occurs as a result of activation in intracellular signal transduction pathways. We think that the evaluation time conditions depend on the factor being tested, taking into account the time lag required for the synthesis and secretion of the protein resulting from the activation of the intracellular signaling pathway.

Recently, it has been demonstrated that the importance of chondrocyte and articular cartilage biomechanics with direct connection to their biochemical functions and activities [34]. In chondrocytes, the cytoskeleton, the cilium, and calcium channels are the main subcellular components involved in the biomechanical response of the cells. In response to mechanical stress, the changes in the mechanical properties of chondrocytes and cartilage matrix may affect the biomechanical behavior of the joint. Therefore, we now plan to study whether changes in the cartilage matrix and chondrocyte properties, as well as passive changes in chondrocyte activity due to mechanical stress, alter their biomechanical behavior, resulting in the change in biomechanical behavior of the joint tissue itself.

Our present study indicates that repetitive compressive loading induces downregulation of mitochondrial ATP production and upregulation of cartilage matrix-degrading enzyme metalloproteinase (MMP)-13 through the NAD-Sirtuin1-Runx2 pathway in OA chondrocytes. Repetitive mechanical stress accelerated the coactivation of NAD-dependent Sirtuin 1-Runx2 in chondrocytes and a resultant increase of MMP-13 production by mitochondria since Runx2 is known to be a promoter of MMP-13 activation. In addition, repetitive compressive loading led to the downregulation of ATP energy generation by mitochondria through a mechanism involving two key mitochondrial regulators, namely the sirtuin 1-NAD interaction, in chondrocytes. We hypothesized that the cell energy regulator Sirtuin 1 and the mitochondrial function regulator NAD, which respond to mechanical stress on cartilage, may control the activity of DNA damage repair and the level of OA-related transcription factor Runx2. These factors may act as protective mechanisms up to a certain threshold load level. However, normal cartilage cells are constantly subjected to mechanical stress. Then, cartilage degeneration progresses when the defense mechanism is no longer able to withstand repeated loading stress and the defense function weakens.

In conclusion, we propose that repetitive compressive loading accelerated the sirtuin activation, which requires and consumes NAD within mitochondria, leading to a decrease of NAD and ultimately in reduced mitochondrial ATP production in chondrocytes. Targeting the mitochondrial NAD-sirtuin 1 pathway may represent a promising avenue for developing innovative therapeutic strategies. Future research should focus on translating these insights into clinically viable treatments to address the underlying mechanisms of mitochondrial dysfunction in OA effectively.

## 4. Materials and Methods

### 4.1. Monolayer Human Chondrocyte Culture

Cultured chondrocytes were isolated and established from human articular cartilage tissues of the non-weight-bearing area of the surgical specimens in patients with OA (disease grade: Kellgren-Laurence grade 3) [10,11,12], where no significant cartilage degeneration was observed at the non-weight bearing areas. Articular cartilage tissues were obtained from the knee joint of OA patients who underwent arthroplastic surgery and who provided written informed consent before participating in the study (female, 60, 63, 67, 71, 73 years old, all 5 patients’ disease grade: Kellgren-Laurence grade 3). The study protocol was carefully reviewed and approved by the University Ethics Committee of St. Marianna University School of Medicine (permission number: 1315). The procedures followed were in accordance with the ethical standards of the Ethics Committee and with the Helsinki Declaration of 1975, as revised in 2000.

The pieces of articular cartilage explants were minced into small pieces, washed with PBS, and digested individually with 1.0 mg/mL collagenase type I (Fujifilm Wako Pure Chemical Inc., Tokyo, Japan) in DMEM (Sigma-Aldrich Co., LLC., St. Louis, MO, USA) for 12 h on a shaking platform at 37 °C. Isolated chondrocytes from each explant were collected following centrifugation, washed four times with PBS, resuspended, and then cultured in DMEM supplemented with 10% heat-inactivated fetal calf serum, 2 mM L-glutamine, and 100 U/mL each of penicillin and streptomycin at 37 °C in a humidified atmosphere of 95% air and 5% CO_2_ as previously reported [33,61]. In the preliminary test, it had been confirmed that there was no significant variation in cellular activities among 5 donors’ chondrocytes derived from non-load bearing areas (all 5 patients’ disease grades: Kellgren-Laurence grade 3). In addition, there was no significant difference in chondrocyte activity of non-weight-bearing areas from 5 OA donors in comparison with that from patients with joint fractures.

### 4.2. Generation of a 3D Cell−Collagen Scaffold Construct for Cultured Human Cells

Three-dimensional cultured tissue was generated from human chondrocytes according to the previously described methods [33,61]. Cultured chondrocytes in a monolayer were collected and resuspended at 1.0 × 10^6^ cells/mL in DMEM. To produce the 3D cell−collagen scaffold construct, the cell suspension was seeded onto a collagen sponge scaffold in a 96-well cell culture plate (AGC TECHNO GLASS Co., Ltd., Shizuoka, Japan) at a density of 1 × 10^5^ cells/scaffold/well and incubated at 37 °C in a humidified atmosphere of 95% air and 5% CO_2_. A porous collagen sponge (5 mm diameter, 3 mm thick) was purchased from AteloCell^®^ (KOKEN Co., Ltd., Tokyo, Japan). The pore diameter averages approximately 100 µm (range from 50 to 200 µm). After the generation of the 3D cell-collagen scaffold construct, the 3D culture tissues were incubated for 12 h in a growth medium for chondrocytes, as appropriate, at 37 °C in a 5% CO_2_ atmosphere.

### 4.3. Repetitive Compressive Loading of the 3D Cell Culture Construct

The 3D cell-collagen scaffolds described above were placed into individual wells of a 12-well culture dish (AGC TECHNO GLASS) and maintained in 1.5 mL DMEM. Then, repetitive compressive loading was applied to the 3D cell-collagen scaffolds with the use of a custom-designed and built apparatus as shown in Figure 1. Repetitive load performances (torque, cycle repeatability, speed) and their adjustability of the load-test apparatus were analyzed after 15 min, 30 min, 1 h, 6 h, 12 h, 24 h, and 72 h of operation, and it was confirmed that their errors at each time point were less than 0.5%.

A physiologic and repetitive compressive loading was applied to the test 3D constructs at 0 kPa (non-loaded) or 25.5 gf/cm^2^ (2.5 kPa) [44,45,46,47].

The repetitive compressive loading experiments were carried out for 15, 60, or 180 min at a rate of 20 cycles/minute in a humidified incubator maintained at 37 °C in a 5% CO_2_ atmosphere. 3D constructs without loading were considered to be the control.

From the results of the preliminary test, for the analysis of protein expression (MMP-13), a 180-min loading condition was added to the 15- and 60-min loading conditions and intracellular/nuclear transcription factor groups were evaluated under 15- and 60-min loading conditions. The evaluation time varied depending on the agent being tested, taking into account the time lag required for protein synthesis and secretion resulting from the activation of intracellular signaling pathways. After the incubation period, culture-conditioned medium and cellular protein were collected and stored at −80 °C till each in vitro assay.

### 4.4. Cellular Protein Content in the 3D Cell-Culture Construct

The total protein content of the cells in each 3D cell-collagen sponge construct was extracted by the following method [45,46,47]. Cell protein contents were collected from the 3D collagen sponges immediately after repeated loading. The 3D cell-collagen scaffolds were minced into small pieces, washed 3 times with PBS, and digested with 1.5 mg/mL collagenase B (Sigma-Aldrich) in DMEM at 37 °C for 12 h on a shaking platform. The isolated cells were centrifuged, washed four times with PBS, and collected for in vitro analyses. Protein samples were collected from the cells for subsequent in vitro assays.

### 4.5. Effects of Repetitive Compressive Loading on Chondrocyte Viability

Effects of repetitive compressive loading on the cell viability (levels of proliferative potential and cell membrane damage) of 3D cultured chondrocytes were analyzed using a Cell Counting Kit (CCK)-8 assay kit and a Cytotoxicity LDH Assay Kit-WST (Dojindo Molecular Technologies Inc., Kumamoto, Japan), respectively, after the treatment of chondrocytes with the cyclic compressive loading or non-loading (control).

### 4.6. Scanning Electron Microscopy

Scanning electron microscopy (SEM) analysis was performed to examine the surface structure of the 3D culture constructs, after the treatment of the 3D cell culture scaffold with the repetitive compressive loading or non-loading (control).

The fixation for SEM analysis was performed in two stages: pre-fixation with an aldehyde fixative and post-fixation with a heavy metal fixative osmium tetroxide. Then, moisture was removed from the sample in advance using ethanol, and the sample was dried by freeze-drying with t-butyl alcohol. The conductive treatment of the sample was performed using the osmium coater method using an SEM detector (FE-SEM/EDS S-4800, 3.0 kV, working distance 9.1 mm, Hitachi High-Tech Co., Tokyo, Japan).

### 4.7. Effects of Repetitive Compressive Loading on Chondrocyte Activity

To verify the involvement of repetitive compressive loading in chondrocyte activity, we analyzed the levels of expression of catabolic factor (MMP-13), after the treatment of chondrocytes with repetitive compressive loading or non-loading (control).

As a measure of chondrocyte catabolic activity in chondrocytes, the levels of MMP-13 production by chondrocytes were studied using ELISA kits (MMP-13 assay kit; Amersham Biosciences, Buckinghamshire, UK) and were measured by an ELISA plate reader (at 450 nm) (Multiskan™ FC Microplate Photometer, Thermo Fisher Scientific Inc., Tokyo, Japan).

### 4.8. Western Blot Assay

To study the effects of repetitive compressive loading on the expression of sirtuin 1 and Runx 2 in chondrocytes, the levels of these two proteins in cells were analyzed by western blotting as previously described [33,59], after the treatment of chondrocytes with mechanical loading or non-loading (control). For western blotting, the antibodies used were a polyclonal antibody against human sirtuin 1 (1:5000 dilution; Abcam Ltd., Cambridge, UK), Runx2 (1:1000 dilution; Cell Signaling Technology, Inc., Danvers, MA, USA), GAPDH/beta-actin (1:10,000 dilution; Proteintech Group, Inc., Rosemont, IL, USA) and the corresponding secondary antibody conjugated with horseradish peroxidase (HRP) [Agilent Technologies, inc., Santa Clara, CA, USA, rabbit IgG for anti-sirtuin 1 antibody (1:5000 dilution), or anti-Rabbit Ig conjugated with (HRP) for anti-Runx2 antibody (1:4000 dilution, Agilent Technologies)]. Seven micrograms of cellular protein were loaded in each lane on SDS-PAGE and 10% of SDS-PAGE was used for the western blotting. The antibody-bound protein bands were visualized, and the densitometry of the signal bands was analyzed using an extended cavity laser system (GE Healthcare Bio-sciences KK, Tokyo, Japan).

### 4.9. Effect of Repetitive Compressive Loading on the Level of NAD in Chondrocytes

To verify whether repetitive compressive loading influences the level of NAD in chondrocytes, concentrations of NAD/NADH in chondrocytes were analyzed by a colorimetric assay (NAD/NADH colorimetric assay kit, Dojindo Molecular Technologies), after the treatment of cells with the cyclic compressive loading or non-loading (control).

### 4.10. ATP Production by Chondrocytes

To study the effect of repetitive compressive loading on the production of ATP by chondrocytes, the level of ATP production of cells was analyzed by ATP assay (Toyo be-net Co., Ltd., Tokyo, Japan), after treatments of cells with cyclic compressive loading or controls.

### 4.11. Statistical Analysis

The results of each experimental condition were determined from the mean of 4 independent experiments. The number of samples in each experiment was *n* = 4. All experimental analyses were performed using GraphPad Prism 10. Data were expressed as means ± standard deviation. For parametric data sets, a one-way analysis of variance (ANOVA) with Bonferroni multiple comparison post-hoc test was used. Non-parametric data sets were analyzed by a Kruskal-Wallis test with Dunn’s multiple comparison post-hoc test. The significance level was set at *p* < 0.05.

## Figures and Tables

**Figure 1 ijms-26-04967-f001:**
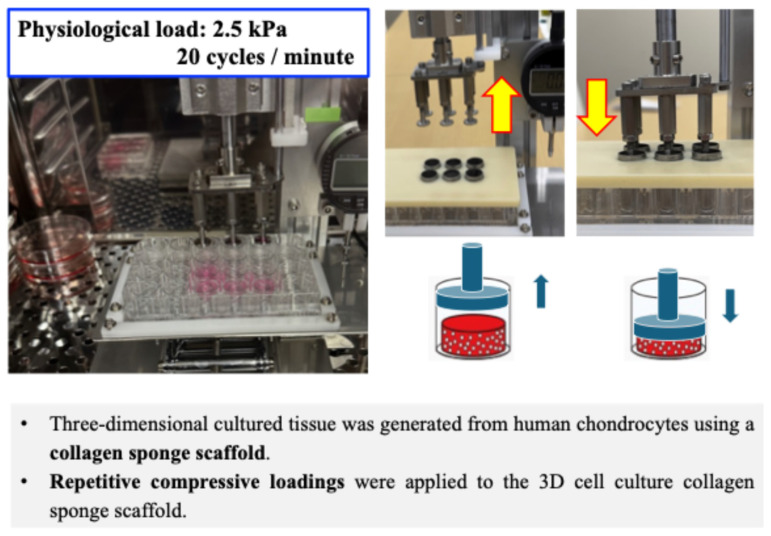
3D cell collagen scaffold construct for human cultured cells. Cultured human cells (chondrocytes) in a monolayer were collected and resuspended at 1.0 × 10^6^ cells/mL in DMEM. The cell suspension was seeded onto a collagen sponge scaffold in a 96-well cell culture plate at a density of 1 × 10^5^ cells/scaffold/well and incubated at 37 °C in a humidified atmosphere of 95% air and 5% CO_2_, to produce the 3D cell collagen scaffold construct. After incubating overnight, one round 3D collagen scaffold disc was placed into each well of a 12-well culture dish. Repetitive compressive loadings were applied to the 3D cell culture collagen sponge scaffold at 0 kPa (non-loaded) or 25.5 gf/cm^2^ (2.5 kPa). (Representative images of generation of the 3D cell–collagen scaffold construct for human cultured cells. Scheme showing mechanical loading of the 3D cell culture construct).

**Figure 2 ijms-26-04967-f002:**
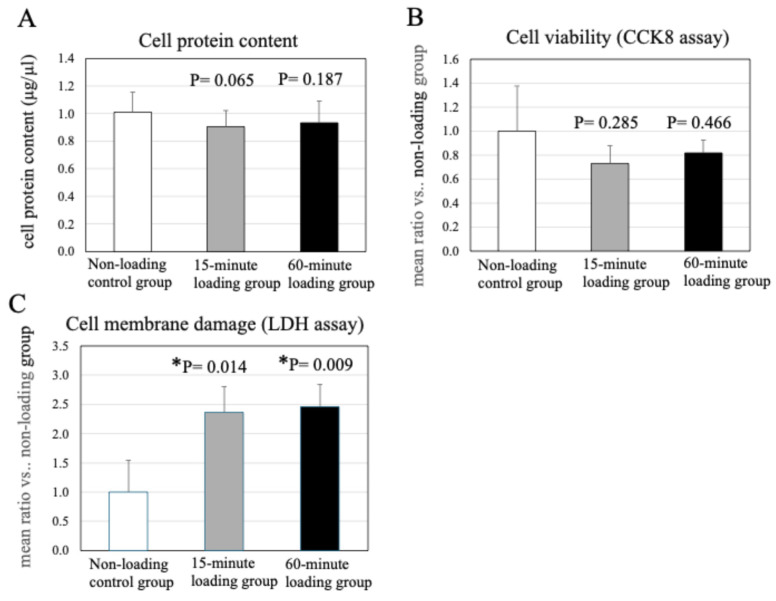
Effects of physiologic and repetitive compressive loading on chondrocyte viability and cell protein content in the 3D cell-culture construct after mechanical loading. (**A**) cell protein content, (**B**) cell viability was analyzed by CCK8 assay), (**C**) cell membrane damage was analyzed by LDH assay.

**Figure 3 ijms-26-04967-f003:**
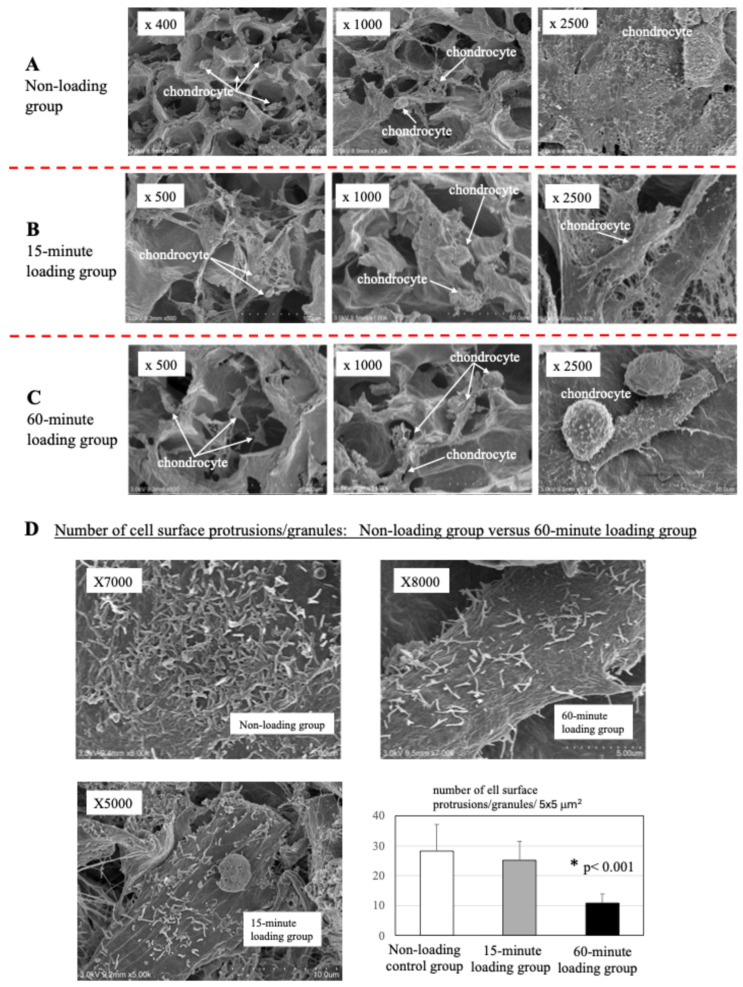
Scanning electron microscopy (SEM) of the 3D cell–collagen scaffold construct. (**A**) Representative SEM images of the 3D chondrocyte collagen scaffold construct (non-loading). (**B**,**C**) Representative SEM images of the 3D osteoblast–collagen scaffold construct (**B**) after 15-min mechanical loading C: after 60-min mechanical loading). (**D**) Number of cell surface protrusions/granules: Non-loading group versus 15- or 60-min loading. Chondrocytes exhibited many protrusions and microgranules on the cell surface. The level of cell surface protrusions/granules was higher in non-loaded chondrocytes than in mechanically 60-min loaded chondrocytes (*p* < 0.001).

**Figure 4 ijms-26-04967-f004:**
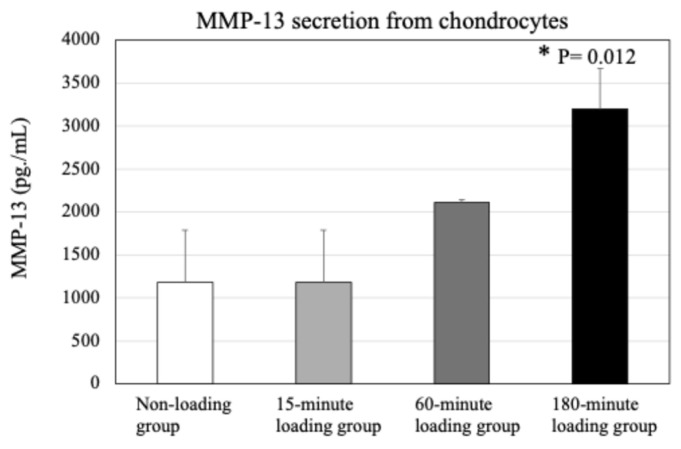
Effects of repetitive compressive loading on chondrocyte activity in the 3D cell-culture construct. The application of repetitive and physiologic mechanical stress for 180 min resulted in significant increases in both MMP-13 and proteoglycan production by chondrocytes in comparison to non-loaded chondrocytes (MMP-13, control vs. 180-min loading: *p* = 0.012. These findings indicate that mechanical stress induced chondrocyte activity.

**Figure 5 ijms-26-04967-f005:**
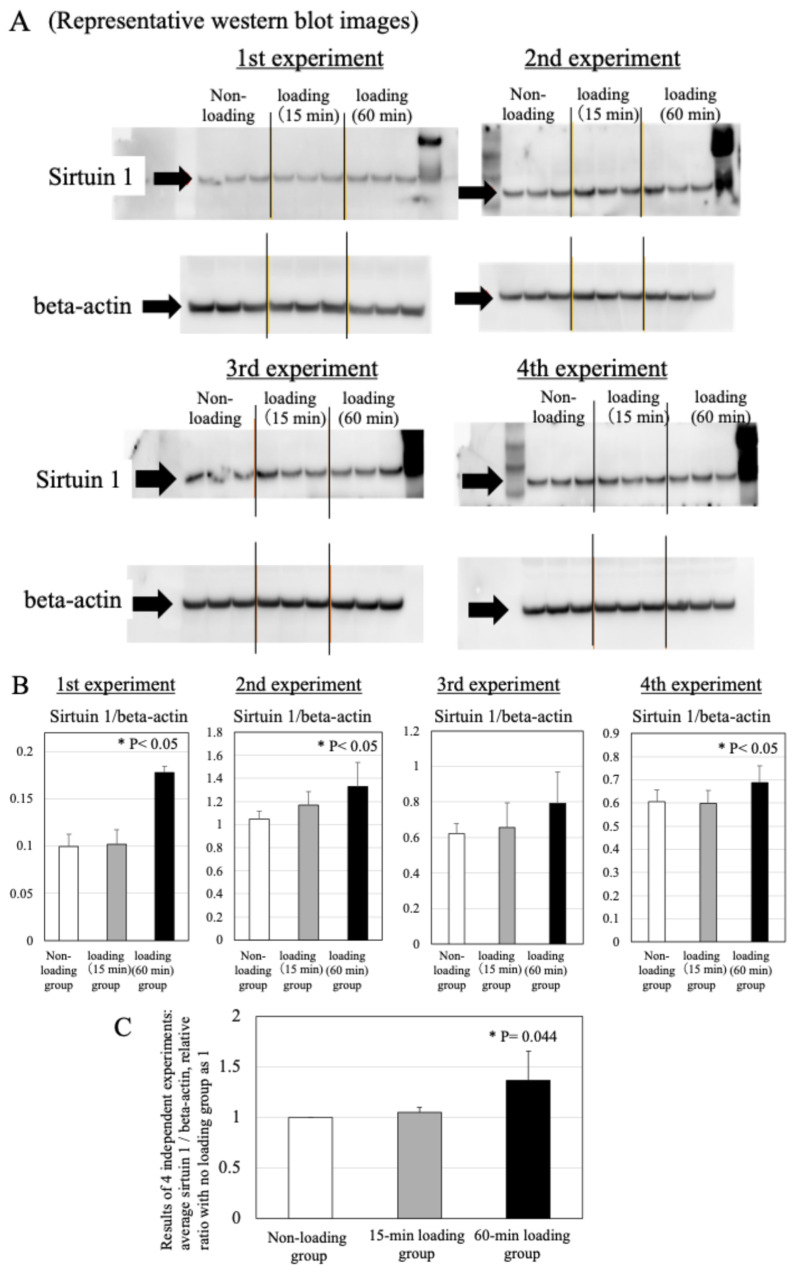
Effects of physiologic and repetitive compressive loading on the expression of sirtuin 1 in chondrocytes. (**A**) Expression of sirtuin 1 (Sirt 1) in non-loaded chondrocytes, 15-min loaded chondrocytes, and 60-min loaded chondrocytes (representative western blotting images: 1st, 2nd, 3rd, 4th experiment). (**B**) Mean ratio relative to beta-actin. Sirtuin 1 protein levels showed a time-dependent increasing trend and were significantly increased in 60-min loaded chondrocytes compared to non-loaded chondrocytes in the 1st, 2nd, and 4th experiments (* *p* < 0.05). (**C**) Results of 4 independent experiments: average sirtuin 1 levels/beta-actin levels, with no loading group set as 1. Treatment for 60 min with repetitive compressive loading significantly upregulated the expression of sirtuin 1 in chondrocytes in comparison with the control (*n* = 4 independent experiments, control vs. 15 min-loading: *p* = 0.164, control vs. 60 min-loading: * *p* = 0.044).

**Figure 6 ijms-26-04967-f006:**
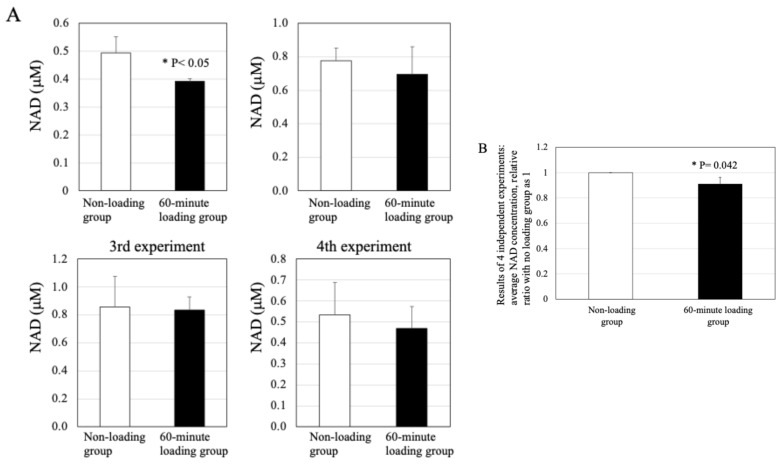
Effect of physiologic and repetitive compressive loading on the concentration of nicotinamide adenine dinucleotide (NAD+) in chondrocytes. (**A**) The application of repetitive and physiologic mechanical stress for 60 min resulted in a decreased concentration of NAD+ in chondrocytes in comparison with the control. (**B**) Results of 4 independent experiments: average NAD concentration (mM), relative ratio with the no loading group set as 1. There was a significant difference in the NAD activity in chondrocytes between the non-loading control and 60-min loading group (*n* = 4 independent experiments, control vs. 60-min-loading: * *p* = 0.042).

**Figure 7 ijms-26-04967-f007:**
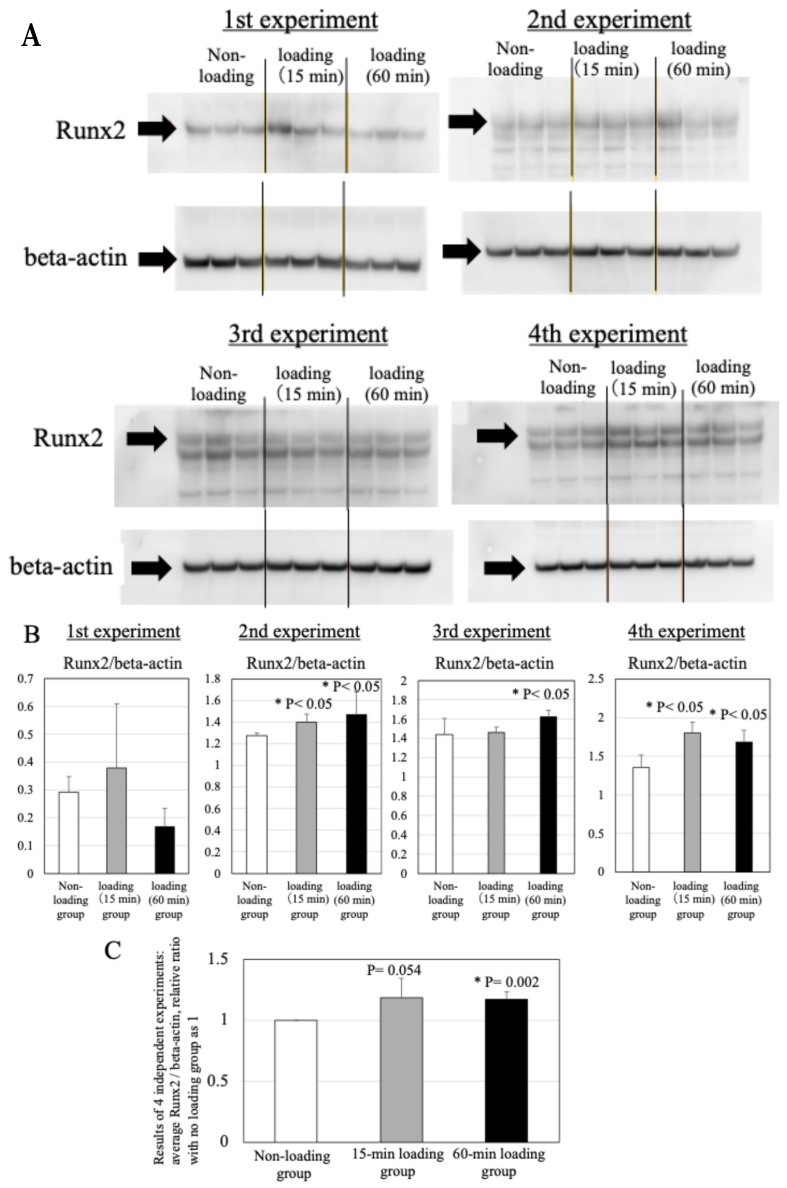
Effects of physiologic and repetitive compressive loading on the expression of Runx2 in chondrocytes. (**A**) Expression of Runx2 in non-loaded chondrocytes, 15-min loaded osteoblasts, and 60-min loaded osteoblasts (representative western blotting images: 1st, 2nd, 3rd, 4th experiment). (**B**): Mean ratio relative to beta-actin. Runx2 protein showed an increase in 15-min and/or 60-min loaded chondrocytes compared to non-loaded chondrocytes in 2nd, 3rd and 4th experiments (* *p* < 0.05). (**C**): Results of 4 independent experiments: average Runx2/beta-actin, relative ratio with the no loading group set as 1. There was a significant difference in Runx2 expression in chondrocytes between the non-loading control and 60-min loading group (*n* = 4 independent experiments, control vs. 15 min-loading: *p* = 0.054, control vs. 60 min-loading: * *p* = 0.002).

**Figure 8 ijms-26-04967-f008:**
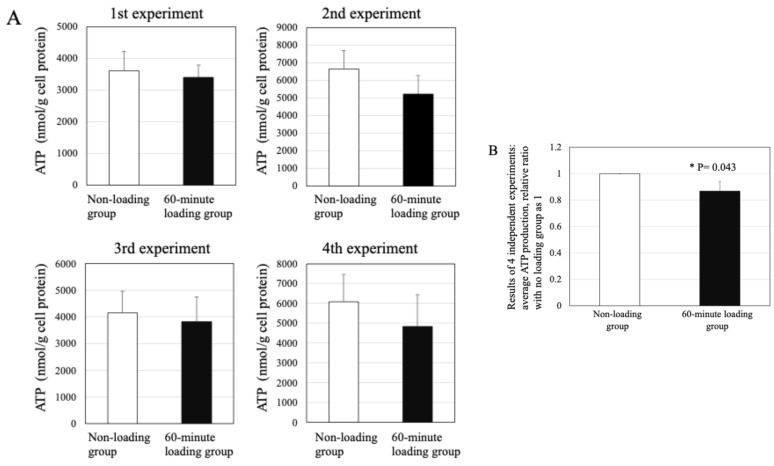
Effects of the physiologic and repetitive compressive loading on the ATP production by chondrocytes. (**A**) The treatment with repetitive and physiologic mechanical stress for 60 min trended to decrease the ATP production in chondrocytes in comparison with the control. (**B**) Treatment with the repetitive compressive loading significantly decreased the levels of ATP production in chondrocytes, in comparison with the control (*n* = 4 independent experiments, control vs. 60 min-loading: * *p* = 0.043).

**Figure 9 ijms-26-04967-f009:**
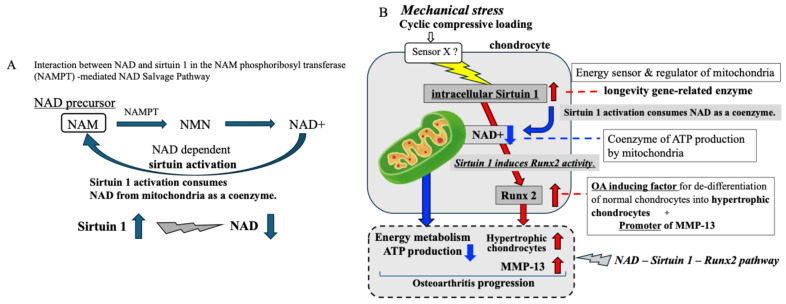
Summary of our current study. (**A**) Interaction between NAD and sirtuin 1 in the NAM phosphoribosyl transferase (NAMPT)-mediated NAD salvage pathway. Sirtuin 1, a longevity gene-related protein, has an important role as a mitochondria regulator and as an energy sensor during ATP production by mitochondria. Sirtuin protein is a NAD-dependent deacetylase, and for its activation, NAD+ is required as a coenzyme. Subsequently, NAD+ is consumed and the level of NAD+ in the mitochondria is decreased. (**B**) Our data indicates that repetitive mechanical stress accelerated the coactivation of NAD-dependent Sirtuin 1 – Runx2 in chondrocytes with a resultant increase in MMP-13 production by mitochondria (since Runx2 is known to be a promoter of MMP-13 activation). In addition, repetitive compressive loading accelerated sirtuin activation, which requires and consumes NAD within mitochondria, leading to a decrease of NAD and ultimately reduced mitochondrial ATP production in chondrocytes. Our findings indicate that repetitive compression loading-mediated mitochondrial dysfunction plays a pivotal role in the progression of OA, primarily by driving downregulation of ATP production and promoting the expression of the matrix-degrading enzyme MMP-13.

## Data Availability

The data that support the findings of this study are available from the corresponding author, (Kazuo Yudoh), upon reasonable request.

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
