# Peer review of "Repetitive Compressive Loading Downregulates Mitochondria Function and Upregulates the Cartilage Matrix Degrading Enzyme MMP-13 Through the Coactivation of NAD-Dependent Sirtuin 1 and Runx2 in Osteoarthritic Chondrocytes"

_ijms, 2025, doi:10.3390/ijms26114967_

Round 1
Reviewer 1 Report
Comments and Suggestions for Authors
My comments are as follows:
1) title is very long.
2) Abstract is too long. Please summarize.
3) Lines 55-56: Introduction on knee OA should be improved. OA is a whole joint disease, involving all joint tissues. Infrapatellar fat pad and meniscus should be added.
4) risk factors for OA are lacking.
5) Lines 71-78: authors explained that cartilage and chondrocytes are subjected to mechanical stress etc, but it should be explained that OA cartilage and chondrocytes changes their biomechanical behaviour (i.e. DOI: 10.3390/biomedicines11071942 etc).
6) Line 95: Please check “[28-.”
7) Line 145: please delete “(AGC TECHNO GLASS)”. This information should be moved to the methods.
8) Lines 149-150: chondrocytes/osteoblasts is unclear. Did authors use osteoblasts in this study?
9) Line 153: CO2 should be corrected.
10) Line 162: cm2 should be corrected.
11) figure 2: It is not clear how long after the compressive loading the cells were collected to measure cell protein content or to evaluate cell viability etc.
12) Line 178: Did authors treat PBMCs? Why? PBMCs are mentioned also in other parts of the manuscript. Please check.
13) Line 180: please check the p-values as it seems that they are inverted.
14) Lines 184-190: this part should be deleted. These results were previously described.
15) Line 200: osteoblasts?? Please check the whole manuscript for consistency.
16) Figure 3: 15 -minute loading group is lacking.
17) section 2.5. : here authors used different conditions that were not characterized in the experiments showed in the previous sections. Why? Authors should show the same conditions in all the experiments. Please correct.
18) “ml” should be “mL”. Please check the whole manuscript.
19) figure 5: western blots should be enlarged.
20) mechanical load is not the only cause of OA. Please revise this.
21) Lines 302-303: The hands are not weight-bearing joints.
22) In the discussion, it should be explained not only that mechanical stress is a contributing factor for OA, but also that tissues in OA changes their mechanical behaviour.
23) limitations of the study should be discussed.
24) Lines 457-458: please add suppliers.
25) Lines 468-469: diameter of pore is not clear. Why is there a wide range?
26) Lines 479-481: all the experiments should be performed using the same conditions: 15, 60 and 180 minutes.
27) section 4.6: There is no mention of whether critical point drying (CPD) was performed. It is not specified whether the samples were coated with a conductive layer. Key SEM imaging parameters are missing, such as the accelerating voltage, the type of detector used etc.
28) authors used a home-made instrument to perform the compressive loading. What is the accuracy/reliability etc?
29) section 4.7: please add plate reader.
30) section 4.8: how many micrograms of protein were loaded on SDS-PAGE? Percentage of SDS-PAGE?
31) section 4.11: software used is lacking.
32) authors should use CRediT – Contributor Role Taxonomy
33) Line 556: 2008? Please check.
Author Response
Reviewer 1
My comments are as follows:
- Comment 1) title is very long.
- Comment 2) Abstract is too long. Please summarize.
Answer to comment 1) and 2)
Thank you very much for your helpful comment. We agree with your comments.
As you comment, we have carefully checked and have shortened the “title” and “abstract” section and all sections without loss of our point. We have revised the title and abstract section in the revised manuscript (please find them in the revised manuscript). Thank you again for your comment.
- Comment 3) Lines 55-56: Introduction on knee OA should be improved. OA is a whole joint disease, involving all joint tissues. Infrapatellar fat pad and meniscus should be added.
Answer:
Thank you very much for your helpful comment. We understand what you mean: "Introduction on knee OA should be improved" and "OA is a whole joint disease, involving all joint tissues. Infrapatellar fat pad and meniscus should be added".
We are sorry that our introduction on knee OA was not enough. Indeed, we can also observe the degeneration and inflammation of soft part tissues, such as infrapatellar fat pad, meniscus and synovial tissues in patients with knee OA. With advance of the disease, the secondary arthritis (such as synovitis, soft tissue inflammation) is also observed in the OA joint tissues.
In the revised manuscript, we have revised the issue in the introduction section (line 49-52).
We appreciate your helpful comment.
- 4) risk factors for OA are lacking.
Answer:
Thank you very much for your valuable comment. As your comment, we have added the information of risk factors for OA in the revised manuscript (introduction section, line 59-67). We appreciate your helpful comment.
- 5) Lines 71-78: authors explained that cartilage and chondrocytes are subjected to mechanical stress etc., but it should be explained that OA cartilage and chondrocytes changes their biomechanical behaviour (i.e. DOI: 10.3390/biomedicines11071942 etc.).
Answer:
Thank you very much for your valuable comment. We agree with your important comment: we should explain that OA cartilage and chondrocytes changes their biomechanical behavior. Thanks to your comment we noticed that. This is very important for us to further understand the pathogenesis and pathophysiology of osteoarthritis.
According to your comment, we have mentioned that OA cartilage matrix and chondrocytes change their biomechanical behavior, as well as the passive changes in chondrocyte activity due to mechanical stress, in the revised manuscript (introduction section, line 111-127).
" In OA, chondrocyte variability and the properties of cartilage matrix components may affect the resulting mechanical behavior of the joint. Recently, it has been demonstrated that the importance of chondrocyte and articular cartilage biomechanics with direct connection to their biochemical functions and activities. Chondrocytes are responsible for the synthesis and degradation of the extracellular matrix/pericellular matrix. The pericellular matrix is considered to be a buffer for physical forces between the chondrocyte and the extracellular matrix. If chondrocytes are subjected to abnormal mechanical stimuli (e.g., excessive loading, joint trauma or malalignment), their metabolism balance becomes altered, causing matrix loss and tissue degeneration, which can lead to OA. In chondrocytes, the cytoskeleton, the cilium and calcium channels are the main subcellular components involved in the biomechanical response of the cells. The changes of mechanical properties of chondrocytes and cartilage matrix may affect their biomechanical behaviour. In pathological condition of OA, chondrocytes and cartilage are subjected to several changes that also modify their biomechanical behaviour. Indeed, OA chondrocytes showed a lower elastic modulus and viscosity compared to healthy chondrocytes. In addition, it has been also demonstrated that OA cartilage acquires a lower Young’s modulus compared to healthy cartilage. Not only does mechanical stress contribute to OA, but the degenerative tissue itself in OA alters the mechanical behavior of the joint."
We appreciate your helpful comment.
- 6) Line 95: Please check “[28-.”
Answer:
We are awfully sorry for our careless mistake. We have revied “[28-29]” in the revised manuscript (line 92). Thank you for your helpful comment.
- 7) Line 145: please delete “(AGC TECHNO GLASS)”. This information should be moved to the methods.
Answer:
Thank you very much for your helpful comment. It was our careless mistake. This information moved to the methods section. We have revised the issue in the revised manuscript (line 158). Thank you again for your comment.
- 8) Lines 149-150: chondrocytes/osteoblasts is unclear. Did authors use osteoblasts in this study?
Answer:
We are awfully sorry for our careless mistake. We did not use "osteoblasts" in the present study. We have deleted “osteoblast” and collected the issue in the revised manuscript (line 163-164). Thank you for your helpful comment.
- 9) Line 153: CO2 should be corrected.
Answer:
We are sorry for our careless mistake. We have revied “CO2” to "CO2"in the revised manuscript (line 166). Thank you for your helpful comment.
- 10) Line 162: cm2 should be corrected.
Answer:
We are sorry for our careless mistake. We have revied “cm2” to "cm2"in the revised manuscript (line 175). Thank you for your helpful comment.
- 11) figure 2: It is not clear how long after the compressive loading the cells were collected to measure cell protein content or to evaluate cell viability etc.
Answer:
Thank you very much for your helpful comment. We agree with your comment.
In all experiments, cell proteins and culture supernatants were collected from the 3D collagen sponges immediately after repeated loading. Since there was no cell culture period after loading stress, we believe that there was no influence of post-cell culture in the further cell growth and the recovery of cellular activity.
As your comment, we have added the sentence and revised the method section in the revised manuscript (line 529-530). We appreciate your helpful comment.
- 12) Line 178: Did authors treat PBMCs? Why? PBMCs are mentioned also in other parts of the manuscript. Please check.
Answer:
We are awfully sorry for our careless mistake. It was our fault. We have deleted "PBMCs" and revised the issue in the revised manuscript (line 191). Thank you again for your helpful comment.
- 13) Line 180: please check the p-values as it seems that they are inverted.
Answer:
We are awfully sorry for our careless mistakes. We have carefully checked all data and have revised the P-values in the revised version (line 193). Thank you again for your helpful comment.
- 14) Lines 184-190: this part should be deleted. These results were previously described.
Answer:
Thank you very much for your helpful comment. We agree with your comment. As your comment, we have deleted the sentences in the revised manuscript (line195-196). We appreciate your helpful comment.
- 15) Line 200: osteoblasts?? Please check the whole manuscript for consistency.
Answer:
We are awfully sorry for our careless mistakes. We have carefully checked our revised manuscript and have revised several mistakes in the revised manuscript (line 210).
Thank you again for your helpful comment.
- 16) Figure 3: 15 -minute loading group is lacking.
Answer:
Thank you very much for your helpful comment. We are awfully sorry that the data of 15 -minute loading group was lacking in Figure 3. It was our fault. Indeed, the results for the 15-minute group were not as significant as those for the 60-minute group. In this 3D cell culture system, it seems that the protein production/secretion from chondrocytes and the change of cell morphology are not significant after 15 minutes of culture, and that 60 minutes of culture may appear to be necessary. In our opinion, in this 3D cell culture system, we think that a 15 minute of culture is not enough for the protein production by the cells and cell morphologic change, and that a 60-minute incubation period is likely required.
We understand that we should show the same conditions in the experiments.
We have added the results of 15-minute loading group and have revised the results, figure legend and Figure 3 in the revised manuscript (line199-206, revised Figure 3, its figure legend line 209-215). Thank you again for your helpful comment.
- 17) section 2.5. : here authors used different conditions that were not characterized in the experiments showed in the previous sections. Why? Authors should show the same conditions in all the experiments. Please correct.
Answer:
Thank you very much for your helpful comment. We agree with your comment.
We performed each experiment under several conditions (incubation times, loading times, etc.) in the preliminary and main experiments.
In the section 2.5, regarding the production of MMP-13 by cultured chondrocytes, although there was a tendency to increase the secretion of MMP-13 enzymatic protein from chondrocytes in the 15-minute and 60-minute groups in comparison with the control, the results for both groups were not as significant as those for the 180-minute group. It is believed that MMP-13 protein synthesis/secretion by chondrocytes may require at least 60 minutes, but about 180 minutes. In the preliminary test, we observed that the amount of MMP-13 production time-dependently increased over time up to about 180 minutes, but when the culture time was 180 minutes or more, the amount of protein produced reached a plateau.
In contrast to the production of MMP-13 enzymatic protein, it has been suggested that changes in the expression and activity of both intranuclear transcription factor Runx2 and sirtuin 1, which are regulatory factors for MMP-13 production by chondrocytes, and mitochondria regulatory factor NAD may appear to occur within 60 minutes, which is shorter than the time required for protein production (MMP-13) by the cells. Therefore, in the present study, for the analyses of sirtuin1 and Runx2, 15-minute and 60-minute incubation conditions were used, and for the analysis of MMP-13 only, a 180-minute incubation condition was additionally added. We wonder if the evaluation time varies depending on the factor being tested, taking into account the time lag required for protein synthesis and secretion resulting from activation of intracellular signaling pathways.
We agree that comment. We have added the data of 15- and 60-minute loading group and have revised the section 2.5 (results, figure legend and Figure 4) in the revised manuscript (line 519-524).
Thank you again for your helpful comment.
- 18) “ml” should be “mL”. Please check the whole manuscript.
Answer:
We have carefully checked our revised manuscript and have revied the" ml" to "mL" in the whole revised manuscript. Thank you again for your helpful comment.
- 19) figure 5: western blots should be enlarged.
Answer:
We appreciate your comment. according to your comments, we have revised the images of western blots in figure 5.
- 21) Lines 302-303: The hands are not weight-bearing joints.
Answer:
Thank you very much for your helpful comment. In the revised manuscript, we have deleted the word "hands". Thank you again for your comment.
- 20) mechanical load is not the only cause of OA. Please revise this.
- 22) In the discussion, it should be explained not only that mechanical stress is a contributing factor for OA, but also that tissues in OA changes their mechanical behavior.
- 23) limitations of the study should be discussed.
Answer to comments 20), 22) and 23):
Thank you for pointing that out to us. We agree with your comment. As you know, OA is a chronic joint disorder recognized as the most common form of arthritis, leading to significant pain, disability, and a reduced quality of life. The progressive OA is primarily marked by the degeneration of articular cartilage, alongside changes in subchondral bone structure and soft part tissues, such as infrapatellar fat pad and meniscus, synovial inflammation, and the formation of bony growths, or osteophytes.
Several key risk factors, including age, genetics, gender, joint injury, obesity, and repetitive mechanical stress influence OA. Aging remains the most significant factor, as cartilage naturally wears down over time. Genetic predispositions increase susceptibility, particularly in hand and knee OA. Obesity also accelerates OA in weight-bearing joints by increasing mechanical load and contributing inflammatory mediators. In addition, joint injuries and occupations involving repetitive motions further stress joints, leading to cartilage breakdown and secondary inflammation of soft part tissues including synovial tissue, infrapatellar fat pad and meniscus. Along with that, metabolic disorders and low bone density also play a role, linking OA with systemic inflammation and joint degeneration.
The various risk factors mentioned above are involved in OA pathogenesis with varying importance. Indeed, mechanical stress is only one of these risk factors. Therefore, to clarify the exact mechanism of OA pathophysiology, it is also necessary to analyze the contribution of factors other than mechanical stress.
In addition, since mechanical stress is thought to affect articular tissues with exception of cartilage tissue, it is necessary to analyze the effects of mechanical stress on the activity of cells excepting chondrocytes.
In the present study, first of all, we have studied the chondrocyte activity, to verify the involvement of repetitive mechanical stress in the OA pathophysiology. Nextly, we are planning to study the effects of repetitive mechanical stress on other cell activities in the joint tissue in vitro.
In addition, we have mentioned that not only does mechanical stress contribute to OA, but the degenerative tissue itself in OA alters the mechanical behavior of the joint, as mentioned above for answer to comment 5.
According to the line which you comment, we have revised the issue mentioned above in the Discussion section in the revised manuscript (line 59-68, 294-307, 437-445). Thank you very much for your helpful comment.
- 24) Lines 457-458: please add suppliers.
Answer:
Thank you very much for your helpful comment. As your comment, we have added the information of 3D collagen sponge cell culture scaffold (KOKEN CO., LTD. Tokyo, Japan) and revised in the revised manuscript (line 502). Thank you again for your comment.
- 25) Lines 468-469: diameter of pore is not clear. Why is there a wide range?
Answer:
Thank you for pointing that out to us. We agree with your comment.
In the present study, we used a porous collagen sponge (5 mm diameter, 3 mm thick, KOKEN CO., LTD. Tokyo, Japan) with a circular porous structure (please see pore structure images below).
According to the product information, the unique porous network allows complete cell and nutrient flow through the pores while providing an increased surface area for cell attachment, proliferation and migration. The collagen sponge is composed of highly purified type I collagen to support cell attachment, proliferation and function. The collagen is lightly cross-linked for mechanical strength and durability, allowing for cell culture. The pore diameter averages approximately 100 microns (ranges from 50-200 microns). In our opinion, we believe that this pore diameter is optimal for three-dimensional culture of chondrocytes in vitro.
< Images from supplier's product information>
We have revised the issue mentioned above in the Method section in the revised manuscript (line 496-497).
- 26) Lines 479-481: all the experiments should be performed using the same conditions: 15, 60 and 180 minutes.
Answer:
Thank you very much for your valuable comment.
We performed each experiment under several conditions (incubation and loading periods, titers of antibodies, etc.). Regarding the production of MMP-13 by cultured chondrocytes, although there was a tendency to increase the secretion of MMP-13 enzymatic protein from chondrocytes in the 15-minute and 60-minute groups in comparison with the control, the results for both groups were not as significant as those for the 180-minute group. We believed that MMP-13 protein synthesis and secretion in chondrocytes may require at least 60 minutes, but about 180 minutes. In preliminary experiment, the amount of MMP-13 production time-dependently increased over time up to about 180 minutes, but when the culture time was 180 minutes or more, the amount of protein produced reached a plateau.
In contrast to the MMP-13 enzymatic protein, it has been suggested that changes in the expression and activity of both intranuclear Runx2 and sirtuin 1, which are regulatory factors for MMP-13 production by chondrocytes, and mitochondrial NAD may appear to occur within 60 minutes, which is shorter than the time required for MMP-13 protein production by the cells. Therefore, for the analysis of protein expression (MMP-13), a 180-minute loading condition was added to the 15- and 60-minute loading conditions, and intracellular/nuclear transcription factor groups were evaluated under 15- and 60-minute loading conditions. The difference in evaluation time depending on the factor examined was due to the consideration of the time lag in protein (MMP-13) synthesis/secretion that occurs as a result of activation in intracellular signal transduction pathways (sirtuin 1-NAD-RUNX2).
We wonder if the evaluation time varies depending on the factor being tested, taking into account the time lag required for protein synthesis and secretion resulting from activation of intracellular signaling pathways.
We are awfully sorry. For NAD, data from the 15-minute loading group was missing, then it was added as supplemental data. (in contrast to the 60-min loading group, no significant difference was observed between the 15-minute loading group and the non-loading control (supplementary data).
Of course, we agree with your comment. We have revised the information of incubation period in the method section and have discussed the issue mentioned above in the revised manuscript (line 419-436, 519-524).
Thank you very much again for your valuable comment.
- 27) section 4.6: There is no mention of whether critical point drying (CPD) was performed. It is not specified whether the samples were coated with a conductive layer. Key SEM imaging parameters are missing, such as the accelerating voltage, the type of detector used etc.
Answer:
Thank you for pointing that out to us. We agree with your comment. We are awfully sorry that important information for SEM analysis was missing.
In the present study, the fixation for SEM analysis was performed in two stages: pre-fixation with an aldehyde fixative and post-fixation with a heavy metal fixative osmium tetroxide. And then, moisture was removed from the sample in advance using ethanol, and the sample was dried by freeze-drying with t-butyl alcohol. As a result of preliminary testing of various conditions and methods, the conductive treatment of the sample was performed using the osmium coater method. It has been confirmed that the osmium coater was able to coat a thin osmium film of only 1 to 2 nm and to deposit evenly even on samples with complex surface structures, making it suitable for high-resolution observation using SEM (FE-SEM/EDS S-4800, 3.0 kV, working distance 9.1 mm, Hitachi High-Tech Co., Tokyo, Japan).
According to your comments, we have revised the method section for SEM analysis as mentioned above. Also, important parameters for SEM imaging, such as the accelerating voltage and the type of detector used (FE-SEM/EDS S-4800, 3.0 kV, working distance 9.1 mm, Hitachi High-Tech Co., Tokyo, Japan), have been added to the Method section (line 545-550).
Thank you again for your helpful comment.
- 28) authors used a home-made instrument to perform the compressive loading. What is the accuracy/reliability etc.?
Answer:
We appreciate your comment. We agree with your comment: we should mention the accuracy/ reliability of our instrument to perform the compressive loading.
For this test, we designed a repetitive load experimental device based on the results of the preliminary test. And then, we had outsourced the production to a medical device manufacturing facility in our university. Repetitive load performances (torque, cycle repeatability, speed) and their adjustability were analyzed after 15 minutes, 30 minutes, 1hour, 6 hours, 12 hours, 24 hours, and 72 hours of operation, and it was confirmed that their errors at each time point were less than 0.5%.
According to the line which you comment, we have added the information of accuracy/reliability of the instrument above mentioned in the revised manuscript (line 510-513). Thank you very much for your comment.
- 29) section 4.7: please add plate reader.
Answer:
Thank you for pointing that out to us. As your comment, we have added the information of a plate reader (Multiskan™ FC Microplate Photometer, Thermo Fisher Scientific Inc., Tokyo, Japan) and revised in the revised manuscript (section 4.7, line 552-553). Thank you again for your comment.
- 30) section 4.8: how many micrograms of protein were loaded on SDS-PAGE? Percentage of SDS-PAGE?
Answer:
Thank you for your helpful comment. We are sorry for the lacking information of blotting. In the present study, first of all, we have performed the pilot studies to determine the experimental conditions in all experiments. In western blotting, preliminary test was carried out to determine the experimental conditions: concentration of loading protein, incubation time, transfer time, percentage of SDS-PAGE, dilution ratio of antibodies, etc. As a result, the most optimal conditions for each experiment were determined. In our current study, 7 microgram of protein was loading in each lane on SDS-PAGE and 10% of SDS-PAGE was used for the western blotting.
In the revised manuscript, we have added and revised the experimental conditions for western blotting for (section 4.8, line 571-572). Thank you again for your comment.
- 31) section 4.11: software used is lacking.
Answer:
Thank you for your helpful comment. We are sorry for our careless mistake. All experimental analyses were performed using GraphPad Prism 10. We have added the information and revised in the revised manuscript (line 587-588). Thank you again for your comment.
- 32) authors should use CRediT – Contributor Role Taxonomy
Answer:
Thank you for your helpful comment. We agree with your comment.
As your comment and according to the "Instructions for Authors", we have revised the information of "Author Contributions" using CRediT Taxonomy in the revised manuscript (line 592-599). Thank you again for your comment.
According to "Instructions for Authors" on IJMS web site
"Conceptualization, M.T. (Masahiro Takemoto) and K.Y. (Kazuo Yudoh); Methodology, M.T. (Masahiro Takemoto), H.F. (Hiroto Fujiya), Y.S., (Yodo Sugishita); Software, M.T., K.Y., and Y.S.; Validation, M.T., H.F., Y.T-S. (Yuki Takahashi-Suzuki), R.F. (Ryuji Fujii); Formal Analysis, H.F., H.N. (Hisateru Niki), K.Y., Y.S.; Investigation, M.T., H.F., H.N., K.Y.; Resources, M.T., K.Y., Y.T-S., Y. S.; Data Curation, M.T., K.Y., Y.T-S., Y. S.; Writing – Original Draft Preparation, M.T., K.Y., Y.T-S., Y. S., R.F.; Writing – Review & Editing, M.T., H.F., H.N., K.Y., Y.T-S., Y. S., R.F.; Visualization, M.T., K.Y., Y. S., R.F.; Supervision, M.T., H.F., H.N., K.Y., Y.T-S., Y. S., R.F. ; Project Administration, H.F., H.N., K.Y.); Funding Acquisition, H.F., H.N., K.Y.”
M.T. (Masahiro Takemoto), H.F. (Hiroto Fujiya), H.N. (Hisateru Niki), K.Y. (Kazuo Yudoh), Y.T-S. (Yuki Takahashi-Suzuki), Y. S. (Yodo Sugishita), R.F. (Ryuji Fujii)
33) Line 556: 2008? Please check.
Answer:
Thank you for pointing that out to us. We are sorry for our careless mistake.
We have revised the year "2008" to "2024" in the revised manuscript (line 605). Thank you again for your comment.
Thank you very much for your valuable review. I would very much appreciate reviewing our revised manuscript and answer letter.

Reviewer 2 Report
Comments and Suggestions for Authors
In this manuscript the authors have investigated the mechanism of mitochondrial dysfunction in OA chondrocytes by observing the effects of subjecting chondrocytes to mechanical stress.
The work is well planned and execute, however there are some issues that need to be addressed:
Line 95: There is a typing error in relation to reference 28
The first obvious question is why did the authors not use chondrocytes from normal individuals as well as OA patients? Given that the cells used were from individuals who already have OA, the authors likely missed key events in the etiology of OA.
Also, for the cells used, the patients should be described in terms of the degree of their osteoarthritic lesions. Depending on the degree of progression of the disease at collection, the experiments performed would likely have varying results due to different degrees of pre-existing damage to the cells.
Were the cells taken from the lesioned areas of the cartilage, or from intact non-load-bearing areas?
In the methods section the authors describe culture of chondrocytes only. However, in the legend for figure 1, they also mention osteoblasts. This point needs to be clarified.
Lines 187-187: Are these part of the legend for figure 2, or are they part of the text? If they are part of the text, then they are repeating information in the previous paragraph and should be removed.
Lines 192 - 194: This text needs to be altered as figure 3 does not quantify cell numbers
Figure 5: The resolution in this figure is not of sufficient quality to allow effective evaluation.
Figure 5 legend: The legend to this figure again describes osteoblasts. What is the relevance of these cells?
Figure 6: It is important for the authors to acknowledge that only one of four experiments showed a significant reduction in NAD; while there certainly seems to be a tendency towards reduction in the other experiments, the authors should consider measures to produce a more robust result.
Figure 6: The resolution in this figure is also sub-optimal. The same for figures 7 and 8.
Figure 7 lacks indicators of statistical significance in part B.
Why did the authors not examine RINX2 expression at the RNA level as well as at the protein level?
4.10: There is yet another confusion about cells: the title suggests the analysis of ATP production by blood cells!
Discussion: The first three paragraphs are superfluous.
Returning to the issue of normal cells; normal chondrocytes are constantly under mechanical stress. Do the results suggest that normal chondrocytes constantly experience reduced ATP production, as well as the other effects described? This should be addressed.
Author Response
Reviewer 2
In this manuscript the authors have investigated the mechanism of mitochondrial dysfunction in OA chondrocytes by observing the effects of subjecting chondrocytes to mechanical stress.
The work is well planned and execute, however there are some issues that need to be addressed:
- Line 95: There is a typing error in relation to reference 28
Answer:
Thank you very much for your helpful comment. We are sorry for careless mistake. In the revised manuscript, we have revised the issue (line 92). Thank you again for your comment.
- The first obvious question is why did the authors not use chondrocytes from normal individuals as well as OA patients? Given that the cells used were from individuals who already have OA, the authors likely missed key events in the etiology of OA.
- Also, for the cells used, the patients should be described in terms of the degree of their osteoarthritic lesions. Depending on the degree of progression of the disease at collection, the experiments performed would likely have varying results due to different degrees of pre-existing damage to the cells.
- Were the cells taken from the lesioned areas of the cartilage, or from intact non-load-bearing areas?
Answers to comment 2, 3, and 4:
Thank you very much for your valuable comments.
In the present study, cultured chondrocytes were isolated and established from the tissues of the non-weight-bearing areas of the surgical specimens of patients with OA (disease grade: Kellgren-Laurence grade 3), where no significant cartilage degeneration was observed at the non-weight bearing areas. In fact, in clinical practice, it was difficult for our university hospital to obtain sufficient amounts of normal chondrocytes for research from healthy individuals (approval for collecting normal cartilage from normal individuals was not obtained from the ethical committee).
A small amount of non-OA chondrocytes was obtained from one young patient with open fracture. Although a sufficient amount was not obtained from this fracture patient's specimen for research, in a preliminary experiment, the cell activity and cell proliferative ability of chondrocytes isolated from OA patients were compared with those from young open fracture patient, and it was confirmed that there was no significant difference between our OA chondrocytes and normal chondrocytes from joint fracture in vitro.
In fact, all OA patients' surgical specimens were carefully collected from the non-weight-bearing areas, and only the areas, where no weight bearing and no cartilage degeneration was observed, were selected to isolate the cells. The characteristics of the isolated and established cultured chondrocytes were compared in a preliminary experiment with the aforementioned chondrocytes derived from young fractures, and it was confirmed that there were no differences in their activities, and then we used them in the current study.
In addition, preliminary test results showed that there was no significant variation in cellular activities among 5 donors' chondrocytes derived from non-bearing areas (all 5 patients' disease grades: Kellgren-Laurence grade 3). In addition, there was no significant difference in chondrocyte activity of non-weight-bearing areas from 5 OA donors in comparison with that from patients with joint fracture. And therefore, the cells were used in this study.
According to the line which you comment (comment 2, 3 and 4), we have added the information of cultured chondrocytes, which were used in the current study, and have revised these issues mentioned above in the revised manuscript (line 470-473, 476, 488-493). Thank you very much for your helpful comments.
- In the methods section the authors describe culture of chondrocytes only. However, in the legend for figure 1, they also mention osteoblasts. This point needs to be clarified.
Answer:
Thank you for your helpful comment. We are awfully sorry for our careless mistakes. We have carefully checked our revised manuscript and have revied several mistakes in the revised manuscript (line163).
Thank you again for your helpful comment.
- Lines 187-187: Are these part of the legend for figure 2, or are they part of the text? If they are part of the text, then they are repeating information in the previous paragraph and should be removed.
Answer:
We are awfully sorry for our careless mistakes. It was our fault. We have carefully checked this paragraph and have clearly revied the issue (we have removed the repeating information) in the revised manuscript (line 195-196). Thank you again for your helpful comment.
- Lines 192 - 194: This text needs to be altered as figure 3 does not quantify cell numbers.
Answer:
We are awfully sorry for our careless mistakes. We have carefully checked all data. Data were corrected by quantifying cell number.
" no significant difference was observed in chondrocyte number in the 3D cell-culture construct between the non-loading control (A) and the mechanical loading groups (B: 15-munute loading) and (C: 60-minute loading) [mean chondrocyte number/100x100 μm2 at x250 image (n=10): (A) 6.2± 2.1, (B) 5.2 ± 1.8, (C): 5.9± 1.8]. "
As your comment, we have revied the issue in the revised manuscript (line 202-203).
Thank you again for your helpful comment.
- Figure 5: The resolution in this figure is not of sufficient quality to allow effective evaluation.
Answer:
Thank you for pointing that out to us. We agree with your comment.
After making sure that the images looked good on the computer screen and when printed out, we have converted the figures and tables to TIFF or JPEG files and inserted them into the manuscript to improve resolution. We have tried to improve the resolution in the figure (western blot images have been enlarged) and proofread the figure in the revised manuscript (Figure 5, 6, 7). Thank you again for your comment.
- Figure 5 legend: The legend to this figure again describes osteoblasts. What is the relevance of these cells?
Answer:
We are awfully sorry for our careless mistake. It was our fault. We have revied the issue of "osteoblasts" in the revised manuscript (line 239). Thank you again for your helpful comment.
- Figure 6: It is important for the authors to acknowledge that only one of four experiments showed a significant reduction in NAD; while there certainly seems to be a tendency towards reduction in the other experiments, the authors should consider measures to produce a more robust result.
Answer:
Thank you for pointing that out to us. We agree with your comments.
In preliminary experiments to determine the loading and incubation conditions in a 3D chondrocyte culture system, we observed a tendency for NAD levels to decrease after cyclic loading. In fact, we recognize that detecting NAD activity in chondrocyte mitochondria, which were extracted from 3D culture, is a technically somewhat difficult experiment. In order to obtain more accurate findings, we believe that it is necessary to master and improve the experimental techniques and increase the number of experiments to eliminate variability in the results.
Therefore, we re-examined the methods and results of multiple previous experiments and have performed new four independent experiments (n= 4 for each experiment) to calibrate the results.
In addition, For NAD, data from the 15-minute loading group was missing, then it was added as supplemental data. (in contrast to the 60-min loading group, no significant difference was observed between the 15-minute loading group and the non-loading control (supplementary data).
If possible, we would like to submit the results of as a supplementary data (please find them in the attached supplementary data file).
Thank you very much for your helpful comment.
- Figure 6: The resolution in this figure is also sub-optimal. The same for figures 7 and 8.
Answer:
Thank you for pointing that out to us. We agree with your comments.
We have tried to improve the resolution in the figures, especially western blot images, and proofread the figures in the revised manuscript (Figure 5, 6, 7, 8). After making sure that the images looked good on the computer screen and when printed out, I converted the figures and tables to TIFF or JPEG files and inserted them into the paper to improve resolution.
Thank you again for your comment.
- Figure 7 lacks indicators of statistical significance in part B.
Answer:
Thank you for your helpful comment. We are awfully sorry for our careless mistakes.
We have carefully checked all data and have added indicators of statistical significance in figures in the revised version (Figure 7B).
Thank you again for your helpful comment.
- Why did the authors not examine RUNX2 expression at the RNA level as well as at the protein level?
Answer:
Thank you very much for your helpful comment. We are very grateful for your comments to further our research.
In the present study, we aimed to investigate whether repeated mechanical loading induces changes in the expression of OA relating-intranuclear factors, sirtuin 1 and Runx2, and mitochondrial function (ATP production, NAD) in chondrocytes and how these changes affect protein expression and cellular energy metabolism (NAD-Sirtuin 1-Runnx2 pathway, Fig. 8B).
In addition to the NAD-Sirtuin 1-Runnx2 pathway, other intracellular factors and mitochondria regulatory factors are also needed to analyze to clarify the exact mechanism of mechanical loading in the pathogenesis of OA. Now, we focus on mitochondrial autophagy (mitophagy) and its relationship to chondrocyte metabolism in OA. For next step of our research project, we plan to investigate in detail the relationship between autophagy-related factors and intracellular signaling pathways, and the effects of weight stress on them. In this project, we are going to examine the levels of mRNA expression of several intracellular factors including sirtuin1, NAD and RUNX2, as well as their protein expressions.
We are awfully sorry, due to financial problem of current research cost constraints current research cost constraints, we would like to evaluate the mRNA analysis collectively as described above.
We hope you understand our opinion. Thank you again for your helpful comment.
- 10: There is yet another confusion about cells: the title suggests the analysis of ATP production by blood cells!
Answer:
We are awfully sorry for our careless mistake. It was our fault. We have revied the issue of "PBMCs" in the revised manuscript (line 581-582). Thank you again for your helpful comment.
- Discussion: The first three paragraphs are superfluous.
Answer:
Thank you very much for your helpful comment. We agree with your comments.
As you comment, we have carefully checked and have deleted first three paragraphs in the discussion section. Thank you again for your comment.
- Returning to the issue of normal cells; normal chondrocytes are constantly under mechanical stress. Do the results suggest that normal chondrocytes constantly experience reduced ATP production, as well as the other effects described? This should be addressed.
Answer:
Thank you for pointing that out to us. We agree with your comments.
As you know, osteoarthritis is a progressive joint degenerative disease caused by the degeneration and wear of articular cartilage, secondary synovitis, and subchondral bone degeneration, osteophyte formation and daily living activities and physical ability decline with age. Regarding OA, in order to develop effective prevention and treatment methods, it is necessary to elucidate the mechanism of mechanical stress perception and response in normal chondrocytes, which is closely related to the cause and pathology of cartilage degeneration. However, there are still many unknowns about how normal chondrocytes sense and respond to mechanical stress and various exogenous stresses that are factors in the onset and pathology of OA, and whether or not they have a defense mechanism (mechanical stress sensing and response mechanism) against pathological excessive stress.
Based on the results of previous research and the results of this study, we hypothesize that the cell energy regulator Sirt-1 and the mitochondrial function regulator NAD, which respond to mechanical stress on cartilage, may control the activity of DNA damage repair (previously reported) and the level of OA related transcription factor Runx2. We think that these factors may act as protective mechanisms up to a certain threshold load level. However, when this defense mechanism (stress resistance) cannot withstand the repeated loading stress or pathological large mechanical stress (once the defense function is reduced), we believe that cartilage degeneration may progress. However, it is still unclear how articular cartilage tissue perceives and responds to mechanical stress through the intracellular signaling pathway. Also, the differences in perception and response activities between physiological mechanical stress and excessive/pathological mechanical stress still remains unknown.
Thus, we believe that these findings will provide clues for developing treatments.
As you point out, normal cartilage cells are constantly subjected to mechanical stress. Based on the results of this experiment, we hypothesize that cartilage degeneration progresses when the defense mechanism (stress resistance) is no longer able to withstand repeated loading stress and the defense function weakens. In the revised manuscript, we have discussed the issue mentioned above, and have revised the discussion section (line 454-460).
Thank you very much for your valuable review. I would very much appreciate reviewing our revised manuscript and answer letter.

Round 2
Reviewer 1 Report
Comments and Suggestions for Authors
No additional comments.
Reviewer 2 Report
Comments and Suggestions for Authors
I have no further comments